# Generating Topologically Consistent BIM Models of Utility Tunnels from Point Clouds

**DOI:** 10.3390/s23146503

**Published:** 2023-07-18

**Authors:** Lei Yang, Fangshuo Zhang, Fan Yang, Peng Qian, Quankai Wang, Yunjie Wu, Keli Wang

**Affiliations:** 1School of Transportation and Civil Engineering, Nantong University, Nantong 226019, China; 2118310009@stmail.ntu.edu.cn (L.Y.); 2018310007@stmail.ntu.edu.cn (Q.W.); wujiewujie0703@163.com (Y.W.); 2School of Geographical Sciences, Nantong University, Nantong 226019, China; 2121110055@stmail.ntu.edu.cn (F.Z.); 2115110199@stmail.ntu.edu.cn (K.W.); 3Key Laboratory of Urban Land Resources Monitoring and Simulation, Ministry of Natural Resources, Shenzhen 518034, China; 4Key Laboratory of Spatial Information Technology R&D and Application, College of Geographic Science, Nantong University, Nantong 226019, China

**Keywords:** point cloud, utility tunnel, scan-to-BIM, hierarchical segmentation, topology reconstruction

## Abstract

The development and utilization of urban underground space is an important way to solve the “great urban disease”. As one of the most important types of urban underground foundations, utility tunnels have become increasingly popular in municipal construction. The investigation of utility tunnels is a general task and three-dimensional laser scanning technology has played a significant role in surveying and data acquisition. However, three-dimensional laser scanning technology suffers from noise and occlusion in narrow congested utility tunnel spaces, and the acquired point clouds are imperfect; hence, errors and redundancies are introduced in the extracted geometric elements. The topology of reconstructed BIM objects cannot be ensured. Therefore, in this study, a hierarchical segmentation method for point clouds and a topology reconstruction method for building information model (BIM) objects in utility tunnels are proposed. The point cloud is segmented into facades, planes, and pipelines hierarchically. An improved mean-shift algorithm is proposed to extract wall line features and a local symmetry-based medial axis extraction algorithm is proposed to extract pipelines from point clouds. A topology reconstruction method that searches for the neighbor information of wall and pipeline centerlines and establishes collinear, perpendicular, and intersecting situations is used to reconstruct a topologically consistent 3D model of a utility tunnel. An experiment on the Guangzhou’s Nansha District dataset successfully reconstructed 24 BIM wall objects and 12 pipelines within the utility tunnel, verifying the efficiency of the method.

## 1. Introduction

With the continuing expansion of urban areas causing issues including traffic congestion, a lack of public services, and the deterioration of the urban environment, the development and exploitation of urban underground spaces is becoming increasingly important. Utility tunnels, which incorporate multiple types of municipal pipelines, are important underground infrastructure in cities for the integrated management and maintenance of pipelines from numerous departments. To maintain the daily operation of utility tunnels, a procedure for investigation and inspection is indispensable [1,2]. Currently, the digitalized maintenance of utility tunnels is still at a low level. Usually, the as-built utility tunnel is different from what was initially designed, which makes maintenance more time-consuming. Accurate building information models (BIM) of the as-built utility tunnels make it more efficient for technical staff to perform maintenance work according to the real situation of the utility tunnels, which is significant for the digitalized management of utility tunnels.

With the development of 3D laser scanners, the cost of laser scanning is declining while the accuracy is improving. Three-dimensional laser scanning has been increasingly used to obtain point cloud data from large building scenes with advantages such as non-contact scanning, wide range, and high accuracy [3,4]. Scan-to-BIM is a common modeling technique that utilizes a 3D laser scanner to capture the point cloud of a building and model it in BIM applications. It is far more efficient than traditional methods of single-point surveying with a total station, which is extensively employed in the architecture, engineering, construction, and facility management (AEC/FM) fields. Presently, various commercial applications or plugins such as Undet and Edgewise [5] offer manual or semi-automatic [6] shape detection and generation of models from point cloud data, with the former for building modeling and the latter for industries with pipes [7]. Regarding utility tunnels with complex internal pipelines and crowded facilities, the existence of occlusions will lead to incomplete data in the scanning results [8]. If existing commercial applications or other methods are used, topological errors can severely affect the quality of the model. Therefore, establishing a precise BIM model for utility tunnels from point cloud data is still a challenging task [9,10].

Several difficulties in using the scan-to-BIM method to model utility tunnels are as follows: (1) the structure of utility tunnels is narrow and packed with intricate interior pipe distributions; (2) segmenting building structures and extracting semantic information of pipes from point clouds with noise and occlusion is challenging; and (3) fitting planes and cylinders based on the extracted clouds may have redundancy and topological errors, making topological relationships of walls and pipe connections difficult to establish. In complicated point cloud environments, available point cloud classification and extraction algorithms currently struggle to reliably identify building structures and segment pipes and their auxiliary facilities.

In this study, a novel method for generating topologically consistent BIM models of utility tunnels is proposed. The main contributions are summarized as follows. (1) A line feature extraction algorithm based on an improved mean-shift and a medial axes extraction algorithm based on local symmetry are proposed to extract parameters of wall and pipeline objects from the hierarchically segmented point clouds. (2) A topology reconstruction method for BIM objects is proposed to recover the topology of wall and pipeline objects through searching and extracting their neighbor information in collinear, perpendicular, and intersecting situations to construct a topologically consistent BIM model.

The remainder of this paper is organized as follows. In Section 2, relevant research on utility tunnel modeling is reviewed. In Section 3, the proposed hierarchical segmentation approach and semi-automatic modeling method are introduced for utility tunnels. In Section 4, the proposed method is applied to a utility tunnel engineering case. In Section 5, the application of the proposed method to the engineering case is discussed. Finally, in Section 6, the experiments are summarized and future work is provided.

## 2. Related Work

The scan-to-BIM approach [11,12] has been widely employed in numerous building modeling [13] applications in the AEC/FM field. In the field of underground space accident search and rescue [14,15,16], relevant research has received increasing attention from researchers. As a significant underground infrastructure, the modeling of utility tunnels can be divided into structure modeling and pipe modeling. Recent research on utility tunnel modeling will be discussed in this section.

Research and applications on building modeling, including the classification of building scenes and the extraction of building structure elements, are currently well-developed. Ai et al. [17] proposed a modeling method based on consistent topological rules, which preprocesses semantic information to obtain evenly structured patches and estimates boundaries using an edge-based primitive extraction method. The model is finally reconstructed to achieve building structure modeling in complex environments. Xiong et al. [18] proposed an automatic modeling algorithm based on unsupervised learning. Core structural components (walls, ceilings, and floors) were recognized and modeled first. Then, openings in the building surface were recognized using a support vector machine (SVM) classifier. Finally, a 3D hole-filling algorithm was used to reconstruct occluded regions not within an opening. The approaches discussed above primarily create planar models of common buildings. However, recognizing structural elements from cloud scenes with noise and occlusions will cause topology errors in the constructed model. As a result, many scholars use the cellular nature of indoor rooms to reconstruct building structures through space partitioning and room reconstruction [19,20,21]. Geometric elements (such as wall lines) are extracted first to construct the cell complex. The graph cuts optimization algorithm is subsequently utilized to recover the building surface geometry. However, regarding the structural modeling of utility tunnels, it is necessary not only to fit surface features but also to construct parametric models based on the wall thickness and height. Furthermore, the interior space of a utility tunnel can be considered an isolated room with a vast length-to-width ratio; thus, building modeling in this area involves problems of fitting and modeling superlong walls.

Another important task in utility tunnel modeling is to model pipelines. Current research on pipeline modeling primarily focuses on pipeline recognition and extraction. Multiple parameters, including orientation, radius, and length [22], must be determined during the process of pipeline recognition and extraction, which varies from the extraction of building planes. Various methods have been presented by scholars for pipeline detection. Random sample consensus (RANSAC) [23,24], Hough transform [25,26,27], Gaussian sphere-based methods, and machine learning methods [28,29] are common approaches that can be used independently or in combination for pipeline shape detection. Proposed by Fischier and Bolles [30], the RANSAC algorithm is widely used in the computer vision (CV) domain. This algorithm calculates the parameters of the target mathematical model based on a set of sample datasets containing outliers, thus obtaining a set of effective data. Schnabel et al. [31] proposed an efficient RANSAC for detecting primitives such as planes, spheres, and cylinders from point clouds. This method begins by generating a hypothetical cylinder from two points and their corresponding normal vectors. Then, the hypothetical cylinder is validated as a cylinder by calculating how many points are on it within the distance and normal-deviation thresholds. However, when repeatedly detecting cylinders, thresholds between different models are needed, and the output of the algorithm depends on the initial selection of the points. The results are occasionally not robust and cannot handle outliers, noise, or incomplete data.

The detection of cylinders in point clouds using the Hough transform [32] comprises two steps: first, identifying the cylinder’s orientation, and second, estimating its radius and position. To identify a cylinder, one point P(u,v) on its axis, one direction vector n(θ,φ), and an estimated radius r are needed. However, computational challenges occur by dealing with an intricate high-dimensional Hough space in this approach. In Ahemd’s work [33], a method that reduces the computational complexity by detecting cylinders in a low-dimensional space is proposed. Furthermore, in Patil’s work [34], the cylinder’s orientation is roughly estimated using a Gaussian sphere at first. Then, the radius is estimated with an improved algebraic circle fitting algorithm. Finally, the connected regions are reconstructed by determining the nearest cylinder relationships. This approach, however, can only detect straight pipes parallel to the orthogonal axis and performs poorly with superlong pipes. The detection results for pipes with elbows are relatively disappointing.

In the process of extracting cylindrical primitives, Gaussian spheres are mainly used to detect the direction of cylindrical pipes. Chaperon et al. [35] proposed a method for extracting cylinders from unstructured point clouds based on the combination of RANSAC and Gaussian spheres. Initially, constraint planes in the Gaussian image are extracted to build a three-dimensional point set with direction. Cylinders are subsequently extracted from the point set. The RANSAC method guarantees the robustness of the extraction result. Liu et al. [1] proposed a cylinder detection method based on Gaussian images for detecting the orientation of pipelines. The problem of rebuilding pipeline equipment in 3D space is simplified into a set of circular detection problems in 2D space. This approach, however, can only extract pipes that are parallel or perpendicular to the ground. Qiu et al. [36] applied a Gaussian sphere to obtain global similarity and detect the axial direction of cylinders. The mean-shift algorithm is used to detect large circles in noise-filled Gaussian spheres. Cylinders are located by fitting 2D circles mapped from points on a plane and the straight pipe segments are smoothly connected. Finally, T-joints and elbows are detected and modeled to restore the connectivity of the pipeline.

Additionally, extracting the pipeline medial axes [6] by transforming the problem of pipeline detection or cylinder recognition into skeleton extraction has been the focus of several researchers. These approaches have superiority in handling scenes with noise and missing point clouds. Liu et al. [37] proposed a BIM automation reconstruction method for full-size complex tubular engineering structures (CTES) based on TLS, which particularly introduces a new algorithm for extracting the medial axis of tubular structures. In the beginning, a region-growing-based extended axis algorithm is utilized to segment the point cloud, and then the geometric parameters of the tubular structure are estimated using a slicing method. Jin [38] used three-dimensional sphere detection to calculate the center point and radius of the cylinder by RANSAC fitting and utilized the points of the curved part of the pipeline to estimate the medial curve at the connection section of the pipeline. Similarly, in Jin’s other work [39], RANSAC was used to estimate the medial axis and radius based on the trajectory of the sphere model, and the medial axis data were checked using principal component analysis (PCA). The outliers in the candidate values of the medial axis were filtered out using a filtering approach. Czerniawski et al. [40] located and extracted the central axis of the pipeline from dense and cluttered point cloud data obtained from a laser scanner by adopting local data grading curvature estimation, clustering, and feature bag matching.

Deep learning approaches for the automatic recognition of pipelines in point cloud scenes are under rapid development as deep learning technology advances. A novel deep learning network for extracting pipes in mechanical, electrical, and plumbing (MEP) systems is presented by Xu et al. [28]. A dataset is established according to the actual engineering situation, but this method has a low classification accuracy for the boundary part of the input data, and the research is still in its early stages, with few digital categories available. Kim et al. [41] also used deep learning approaches for the automatic identification of pipelines in their study. A pipeline and automatic elbow-recognition method is proposed based on curvature information and convolutional neural network primitive classification. This method enables the recognition of simple and S-shaped elbows. This approach, however, is insufficiently accurate when detecting lengthy pipeline sections and is strongly influenced by crowded scenes, making it difficult to handle complex long pipeline sections with elbows in utility tunnels. Moreover, labeling and training data are costly in that a considerable number of samples must be labeled as training sets.

The pipeline modeling approaches described above are mostly intended for pipes with short lengths, few elbows, or no elbows at all. Currently, few studies regarding the approaches for dealing with pipe connections are presented, particularly in utility tunnels. However, it is typical that superlong pipes with many elbows exist in utility tunnel scenes. If extracted directly from the point cloud of a utility tunnel with the present methods, topological errors will affect the quality of the model, which is an urgent problem.

Research on urban building modeling takes the lead in the present studies on scan-to-BIM, while research on utility tunnel modeling is not as popular. During the modeling process of utility tunnels, two categories of objects are created: building structures (wall facades, floors, ceilings, etc.) and pipes. Building structure modeling is comparable to modeling common buildings, and extensive studies have been conducted on the classification of point cloud scenes and the extraction of structural elements. The majority of pipeline modeling research focuses on the identification and extraction of pipe cylinders. It is a comprehensive issue that pipes cannot be explored as an isolated point cloud in utility tunnels but must be considered alongside various types of pipelines and auxiliary facilities in utility tunnels. In addition, many difficulties still exist in the connection of utility tunnel building structures, as well as the identification, segmentation, and extraction of superlong pipes. Therefore, hierarchical segmentation and accurate construction of the topological relationship among building elements are needed. As a result, a hierarchical segmentation method is presented in this paper for the unsupervised recognition and extraction of various elements in utility tunnels. Another topology reconstruction method is presented to avoid probable redundancies and topological errors in the procedure of generating BIM models.

## 3. Proposed Methods

The suggested method for generating topologically consistent BIM models of utility tunnels is depicted in Figure 1, which comprises five major steps: (1) point cloud preprocessing, including the segmentation and filtering of point clouds; (2) hierarchical segmentation of point clouds (in this step, building structures containing walls, ceilings, floors, and pipe systems are segmented based on the calculation of point cloud perpendicularity using a threshold method); (3) extraction of building structures and pipe parameters, geometric parameters such as wall lines, wall heights, and pipe axes; (4) topology reconstruction, which reconstructs the topological relationship of the building object by fusing segmented wall lines and pipe axes; and (5) BIM object reconstruction, which includes reconstructing the BIM model of the utility tunnel using the topologically reconstructed wall lines and pipe axes.

### 3.1. Preprocessing

The raw point cloud data P have been precisely registered. Segmenting and filtering point clouds are two aspects of point cloud preprocessing. Considering the large length-to-width ratio and irregular outer contour of the utility tunnel, as well as the occlusion and incomplete point cloud data of the internal pipe system, a large amount of time is required if semantic segmentation and shape identification are carried out on the entire point cloud. Thus, as illustrated in Figure 2, partitioning the utility tunnel’s point cloud data into blocks is an effective strategy.

Point cloud filtering is the other task of point cloud preprocessing. Outliers may be included in the scanning results during the 3D laser scanning process due to the surrounding environment and instrument-related reasons. Outliers not only degrade the quality of the whole point cloud but also impair local point cloud feature estimation, resulting in inaccurate computation outputs. To filter and denoise the point cloud data in this paper, a statistical outlier removal approach (SOR) is applied. This method performs statistical analysis on each point in the point cloud’s neighborhood, and outliers that do not adhere to the preset statistical distribution are deleted. A point number threshold is set according to a statistical model to check all points in the point cloud data. Points that fail to meet the threshold are removed as outliers.

### 3.2. Hierarchical Segmentation of the Point Cloud

The internal structure of the integrated utility tunnel is quite complicated, and the objects that must be built in its BIM modeling process typically include wall facades, floors, ceilings, pipes, and their auxiliary facilities, which need to be retrieved from the point cloud. Many researchers have included point cloud pre-segmentation in their pipe extraction research, but the classification of pipes in utility tunnels remains a difficult topic. In this paper, a hierarchical segmentation method for extracting the building elements required for BIM modeling from a point cloud is presented. In the beginning, the perpendicularity of the point cloud blocks is calculated. Then, threshold segmentation is utilized to distinguish facades and planes based on the perpendicularity calculation results, which are identified as wall facades, floors and ceilings. Finally, the floors and ceilings are separated from the segmented point cloud blocks using the slicing approach. As illustrated in Figure 3, after hierarchical segmentation, wall facades, floors, ceilings, and the pipe system are precisely separated, with wall facades, floors and ceilings meeting the needs of further modeling of building structures. The remaining point cloud of pipes and their auxiliary facilities is further used for reconstructing the pipe system.

### 3.3. Extraction of Structure Parameters

The point cloud of wall facades obtained through hierarchical segmentation is subsequently used to extract wall line features and height information. The wall lines, height, and thickness are three critical parameters for generating a precise BIM model of walls. Among them, wall lines and height cannot be directly obtained and need to be extracted from the point cloud. After horizontally slicing the point cloud blocks, wall line features are retrieved with the mean-shift algorithm. The process of slicing the point cloud blocks is shown in Figure 4. Accurate wall point clouds are obtained by selecting proper elevations and thicknesses to slice. Line features are extracted according to the slice afterward. These wall line features are made up of several short line segments that need to be restored to complete wall lines.

To estimate the height of the building facade, a clustering algorithm is employed. The principle is to identify and cluster point clouds of different elevations (classified along the *Z*-axis direction). The clustering results are used to generate the elevation distribution histogram. The height of the building facade is the absolute distance between the peak values in the histogram (as shown in Figure 5).

### 3.4. Extraction of the Pipe Parameters

#### 3.4.1. Medial Axis Point Set Extraction

The extraction of the medial axis point set of the pipeline is discussed in this section. Based on the symmetry of cylinders, a method of mirror reflection is proposed to extract the medial axis point of the pipeline from the point cloud. Regarding a certain point cloud $P_{pipe}$ of a pipe, if a point *p* and its nearest neighbor *N* are in the point cloud, the neighbor point $P_{N}$ has symmetry. As shown in Figure 6, for each point $P_{i}^{N}$ in $P_{N}$, the mirror reflection corresponding relationship can be acquired by reflecting it on the mirror plane and finding the nearest neighbor within a fixed radius. After determining the mirror reflection’s corresponding relationship, the Levenberg-Marquardt (LM) algorithm (Formula (1)) is used to minimize the distance between the mirror reflected points and their corresponding points and optimize the original symmetrical plane.
(1)$$S^{\mathrm{refl}}=\arg\min\sum_{i=1}^{N_{p}}d(S^{refl}(p_{i}),q_{i})$$
where $S^{refl}$ is the mirror reflection plane, represented by the point *p* and the normal vector ***n***, and the reflection plane is calculated through PCA. Points $P_{i}$ and $P_{i}^{\prime }$ are a pair of symmetric points near the mirror reflection plane *S* in the point cloud.

An accurate symmetrical plane can be obtained using the mirror reflection symmetry approach, and the extraction of the pipe medial axis point set is then considered. The direction of the pipe is determined by the intersection point direction of the symmetrical plane with the nearest point set. The point cloud of the pipe cross-section is obtained based on this direction, and the RANSAC circle fitting method is subsequently used to calculate the axis point set of the pipeline. As shown in Figure 7, the medial axis point set of the entire pipe can be obtained after repeated calculations of the circle fitting approach. Finally, filtering methods are used to denoise and remove outliers.

#### 3.4.2. Medial Axis Polyline Extraction

After obtaining the medial axis point set of the pipeline, the next step is to extract the pipeline medial axis line. In this section, an improved mean-shift algorithm is used to refine the medial axis point set, and RANSAC is used to extract the axis line. First, the axis point set is randomly sampled with 10% of the input points chosen as initial samples to reduce computational complexity. The sampling points are skewed towards the maximum point density region. The conventional mean-shift algorithm is then enhanced by applying the normalization formula [42,43] (Formulas (2)–(4)) to extract the medial axis point set of pipes from the original point cloud and modify the sample points of the original point cloud.
(2)$$x_{i}{}^{k+1}=\frac{\sum_{j\in J}P_{j}{}^{k}\alpha_{\mathrm{ij}}{}^{k}}{\sum_{j\in J}\alpha_{\mathrm{ij}}{}^{k}}+\lambda \frac{\sum_{i\prime \in I\backslash \{i\}}(x_{i}{}^{k}-x_{i\prime }{}^{k})\beta_{ii\prime }{}^{k}}{\sum_{i\prime \in I\backslash \{i\}}\beta_{ii\prime }{}^{k}}$$
(3)$$\alpha_{\mathrm{ij}}=\frac{\theta (\left\|x_{i}-p_{j}\right\|)}{\left\|x_{i}-q_{j}\right\|},\beta_{ii\prime }=\frac{\theta (\left\|x_{i}-x_{i\prime }\right\|)}{{\left\|x_{i}-x_{i\prime }\right\|}^{2}}$$
(4)$$\theta (\left\|x_{i}-p_{j}\right\|)=e^{-{\left\|x_{i}-p_{j}\right\|}^{2}/R^{2}},\lambda =\mu \sigma (x_{i}{}^{k})$$

The classical mean shift algorithm is represented by the first term in Formula (2). The second term is a regularization term that prevents the refined sampled points from excessively clustering when they shrink to their local central positions. λ is the equilibrium constant between the two terms and it is usually set to the empirical value μ = 0.35. The RANSAC method is used to recover the pipeline medial axis after point cloud refinement. First, two points $A_{1}$ and $A_{2}$ are chosen at random from the axis point set to generate a hypothetical straight line. Then, the number of remaining points that fall on the hypothetical straight line is examined to determine if it meets the predetermined criteria. The hypothesis is regarded as reliable if it fits the pre-set threshold. Following a series of random tests, the parameters of the hypothetical straight line with the greatest number of supporting points are selected as the parameters of the pipe medial axis. This process is repeated for each point cloud block to obtain the pipeline axis line set $L_{P}$. Finally, each medial axis line li in the medial axis line set $L_{P}$ is smoothed. The radius of the pipe is determined by the average radius of all fitted circles on the medial axis line. In this section, the parameters of orientation and radius that are necessary for the reconstruction of pipes are obtained.

### 3.5. Topology Reconstruction

We acquired a preliminary wall line segment set L and a pipeline axis line segment set $L_{P}$ in Section 3.3 and Section 3.4. In this section, the topological relationships of these segments are reconstructed by merging them into continuous polylines. In Sets L and $L_{P}$ and their adjacent line segments, there are three spatial relationships between the line segments, as illustrated in Figure 8: collinear, perpendicular, and intersecting. We propose a method for fusing of collinear line segments and merging intersecting line segments based on line feature sampling and neighbor search (perpendicularity is a special case of intersection). The suggested method is introduced below, taking the topological reconstruction of wall lines as an example. The topological reconstruction procedures of the medial axis of the pipe are comparable.

The first step is to search for neighboring line segments. Above all, the line segments in set L are sampled uniformly with a pre-set sampling interval to produce a set of sampling points $P_{s}=\left\{P_{s1},P_{s2},\cdots ,P_{sn}\right\}$. Then, a data structure $S_{i}=\left(x_{i},y_{i},z_{i},L_{i}\right)$ is constructed based on the sampling point set $P_{s}$ and the line segments to which they belong, where $x_{i}$, $y_{i}$, and $z_{i}$ represent the spatial position of the sampling point, and $L_{i}$ represents the ID of the line segment from the set of the wall line segments L to which the sampling point $S_{i}$ belongs. Then, a KD-tree is constructed for the point-line set S, and for each endpoint $l_{1}$ and $l_{2}$ of each line segment, neighboring points $S_{j}$ within a radius R are searched, and neighboring line segments are identified based on the ID of $S_{j}$. This process is iterated until all endpoints of all line segments are traversed, obtaining the neighborhood Mi of each line segment, which constitutes the set of neighboring domains $M=\left\{M_{1},M_{2},\cdots ,M_{n}\right\}$.

The second step is to fuse collinear line segments. First, each line segment $L_{i}$ in set L and its neighboring line segments $L_{j}$ are traversed to determine if $L_{j}$ is parallel to $L_{i}$. The longer of the two collinear line segments is selected as the direction for the fused line segment $L_{new}$. Next, the neighboring line segment and $L_{i}$ are projected onto the chosen direction $L_{new}$, and the coordinates of $L_{i}$’s endpoints are updated accordingly. The information from $L_{j}$ is added to $L_{i}$, and the information related to the neighboring line segment $L_{j}$ is removed from $L_{i}$. Then, the information of all other line segments except for $L_{i}$ is updated, and the set of the wall line segments L is updated accordingly. Finally, this process is repeatedly performed until all collinear line segments have been fused, and the updated wall line segment set is denoted by $L^{\prime }$.

The last step is to merge intersecting line segments. First, each line segment in the set $L^{\prime }$ and its neighbors is traversed. If the neighboring line segments form a certain angle with the current line segment, the intersection point O of the neighboring line segment and the current line segment are determined. The position relationships between the intersection point O and the current line segment are determined in three cases: on the line segment, on the extended line, or on the reverse extended line. If the current line segment has multiple neighboring line segments, the distance value $D=\left\{{\Delta D}_{1},{\Delta D}_{2},\cdots ,{\Delta D}_{n}\right\}$ from each intersection point $O_{i}$ to the endpoint of the current line segment is calculated, and the elements in D are sorted. The distance adjustment value $min\left\{D\right\}$ is selected as the endpoint of the line segment. Then, the endpoint coordinates of the current line segment are updated based on the distance adjustment value. Finally, all line segments in $L^{\prime }$ are traversed until all line segments are updated to merge the intersecting lines. A schematic illustration of the topological reconstruction of the wall line features and pipe medial axes is shown in Figure 9.

### 3.6. Reconstruction of the BIM Objects

Using the aforementioned method, relevant parameters are obtained for generating BIM structures and pipe objects. In this stage, a Revit plugin for semi-automatically generating BIM objects within a utility tunnel is developed.

Solid wall objects are created for modeling building structures based on the user-input wall thickness, the extracted wall line features, and the wall height. The WallCreate function provided by Revit is used to generate BIM solid wall objects. Rebuilding BIM walls relies on the elevation, wall type, and thickness parameters. Gaussian clustering is utilized to extract the elevations H by sorting the bottom elevations of walls. The wall type is determined by the wall material and thickness. To create BIM solid wall objects, the proper elevation, wall type, and wall thickness are chosen for each wall line. Topological connection relationships between walls are established, and reference information is appended. Table 1 shows some examples of the structure and relevant parameters.

The PipeCreate function provided by Revit requires six major parameters when modeling pipe systems: the type of pipe system, pipe type, elevation, pipe diameter, and the starting and ending points of the pipe. Element filters are used to select the target pipe system type and pipe type. The pipes’ elevation Hp is calculated using Gaussian clustering similarly. To generate a BIM pipe object, the appropriate pipe system type, pipe type, elevation, and diameter are selected for each pipe medial axis. The topological connection relationships between pipe segments are established, and connectors are added to the elbows’ positions (as shown in Figure 10). Notably, the pipe model is generated automatically in the Revit mechanical template and then needs to be linked to the Revit architectural template in which the model of the building structure lies. The semi-automatic modeling of the utility tunnel is now finished. The structural features and relationships that comprise the utility tunnel are depicted in Figure 11.

## 4. Case Study

To verify the feasibility of the semi-automatic scan-to-BIM method proposed in this paper for modeling urban utility tunnels, an experiment is performed on the automatic modeling of the utility tunnel under Pearl Bay Plaza in Nansha District, Guangzhou City.

### 4.1. Project Overview

Pearl Bay Plaza is located in the starting area of Pearl Bay in Nansha District, Guangzhou City, Guangdong Province, on the tip of Lingshan Island (Science Island) (117.17° E, 31.89° N). The nearby underground utility tunnel is approximately 2.3 km long and comprises municipal pipeline channels for electric power, cables, fire protection, water supply, and drainage (Figure 12). The utility tunnel located underground at Pearl Bay Plaza is adjacent to the sunken square. The outline of the utility tunnel is an irregular polygon with a length of 108 m, a maximum width of 5 m, and a building height of 2.5 m. The pipes are stacked in tiers along the wall, with a maintenance route in the center. A total of 91 scanning stations were set up for the scanning task of Pearl Bay Plaza, with a total of 1,244,204,895 scan points. The point cloud data of the utility tunnel, shown in Figure 13, consisted of 20 scanning stations with a total of 227,708,796 scan points. The hardware environment used for the experiment was an Intel Core i7-11800H processor with a frequency of 2.3 Hz and 16 GB of RAM.

### 4.2. Preprocessing

Trimble scanners were used to conduct 3D laser scanning of the sunken plaza, underground utility tunnel, and other areas of the Pearl Bay Plaza. The original point cloud was acquired after a set of preprocessing procedures such as registration, which clearly highlighted the main structures within the utility tunnel, enabling the accurate extraction of building geometry parts. The raw point cloud data were converted to E57 format before preprocessing.

The point cloud of the utility tunnel is partitioned into four blocks based on the building structural characteristics (Figure 14) in the first step of preprocessing. The related information about the parameters for each block is presented in Table 2. Finally, the SOR approach is used to filter and denoise outliers in each block.

### 4.3. Hierarchical Segmentation of the Point Cloud

The perpendicularity of the point cloud blocks is calculated in the first step of hierarchical segmentation of the point cloud, and then the blocks are partitioned into facades and planes based on the results of the perpendicularity calculation. Facades and planes are distinguished by coloring. The second step is threshold segmentation, where point clouds representing wall facades are extracted for wall reconstruction by adjusting parameters, while planes are identified as the floors and ceilings of the utility tunnel, and the point clouds of the pipeline system serve as the basis for subsequent work on pipe recognition. In this manner, the building structures of the utility tunnel are accurately and completely extracted from the point cloud. Figure 15 shows the results of the hierarchical segmentation of three blocks, represented by three different colors corresponding to wall surfaces, floors, and piping systems. The blank spots on the wall surfaces represent occlusion regions.

### 4.4. Extraction of the Structure Parameters

Figure 16 exhibits the utility tunnel’s outer wall line features that are extracted from the sliced point cloud blocks using the principal component analysis approach, yielding a collection of 42 segments that define the wall boundary and are denoted as L. At this time, the wall’s topological relationship has not been properly reconstructed, and gaps still exist between the line segments, making it incapable of generating an entire BIM solid wall object directly. The clustering method, which relies on the distribution of points at different elevations to identify the elevation of the facade, is used to extract the height of the wall.

### 4.5. Extraction of the Pipe Parameters

The process of extracting the medial axis of the pipeline is shown in Figure 17. The blue part represents the point cloud of a 250 mm diameter water pipe, with a central elevation of 240 mm, a total length of 112.7 m, and multiple elbows. First, the pipeline is segmented at positions selected in the straight sections for topological reconstruction, and then the medial axis points of each segment are calculated and fitted with a circle using the RANSAC method. The radii of all the fitted circles are equal since the inner diameter of the water pipe is uniform without any fluctuation zone. Finally, the medial point set is refined using the improved mean-shift algorithm, and the medial axis of the pipeline is fitted using the RANSAC method that result in a set of medial axis segments, denoted as $L_{p}$.

### 4.6. Topology Reconstruction

After obtaining the roughly extracted set of wall line segments L and the set of pipe medial axis segments $L_{p}$, the proposed method was employed for a topological reconstruction experiment based on the line feature nearest-neighbor sampling search method, including the fusion of collinear line segments and the merging of intersecting lines. Figure 18 shows the result of wall topological reconstruction, where all 22 straight line segments in the Set L are connected to their neighboring line segments to form a closed polyline. Figure 19 shows the result of medial axis topological reconstruction, where the cable pipes on the north side of the utility tunnel and the water pipes on the south side are represented by polyline segments in the same manner, with the former consisting of 3 straight line segments and the latter consisting of 12 straight line segments.

### 4.7. Reconstruction of the BIM Objects

After organizing the extracted wall features and pipe medial axes, an external program is developed for semi-automatic BIM modeling based on the Revit API. The program, implemented in C# and developed in the Visual Studio 2022 environment, turned the utility tunnel’s point cloud data into a parameterized BIM model using the extracted geometric information.

The wall type in this experiment is selected as “conventional 200 mm”, and other relevant parameters are shown in Table 3. Twenty-four solid wall objects are generated by reading information of the outer wall line using the developed external program, totaling 223 m in length. Since the interior of the utility tunnel is an isolated place, inner walls are not required to be generated. Finally, the floor and ceiling are placed to complete the reconstruction of the utility tunnel building structure.

Regarding the modeling of pipes, it is essential to mention that there are three types of pipes in the case study (illustrated in Table 4), which are ordinary water pipes (in blue), fire water pipes (in red), and cable pipes (in green), with quantities of 1, 3, and 8, respectively. The diameter of the water pipe is 250 mm; the fire pipe is divided into a main pipe with a diameter of 200 mm and a branch pipe with a diameter of 150 mm; all 8 cable pipes have a diameter of 100 mm and are arranged with 4 cables at two different elevations. Finally, the pipe brackets are fixed on the wall by means of the self-built family.

After completing the building and pipe system modeling, the BIM model of the pipes will be linked to the model building structure, forming a complete BIM model of the utility tunnel (Figure 20).

## 5. Discussion

In the experiment above, the utility tunnel under Pearl Bay Plaza is reconstructed from a point cloud to a parametric BIM object. The experiment on the reconstruction of the utility tunnel is based on the reconstruction of building and pipe objects, and some issues encountered in the experiment are discussed in this section. First, the utility tunnel has a long horizontal depth but is relatively narrow perpendicularly. Analyzing the entire point cloud can be time-consuming. Thus, it is preferable to partition it into blocks and merge them at the end of the experiment. Second, the result of wall line extraction is directly affected by the thickness of the sliced point cloud and the position of the slices when employing the slicing method after hierarchical segmentation. The slices should prevent occlusion areas and have sufficient thickness (0.3 m in the experiment) to ensure that the sliced point cloud is capable of completely representing the wall facade. Third, due to human or natural factors, slight slopes exist in the walls and pipes throughout the utility tunnel construction process. The Gaussian clustering method was employed in the experiment to identify the elevation of the walls and pipes, ignoring the existence of such slopes. In addition, several BIM wall elements could not be connected where the corner place was located, and certain pipe objects could not be connected to the following pipe section due to their angles. Further improvement of the code for reconstructing BIM objects is required to tackle the difficulty of connecting BIM elements. Finally, detailed parts of the reconstructed pipes, such as valves, instruments, and flanges are not considered in the experiment. Only BIM objects of the pipes and brackets are reconstructed.

The experiment on the reconstruction of the utility tunnel BIM model demonstrates that the hierarchical segmentation method is capable of accurately separating basic architectural elements such as facades and planes in complicated point cloud sceneries. The identification results of wall line segments and pipe centerline segments are outstanding on this basis. The line feature sampling neighbor search approach is utilized to connect collinear and intersecting line segments to reconstruct the topological relationships of wall lines and pipe medial axes, which meets the demand of BIM object reconstruction.

## 6. Conclusions

A novel method for generating topologically consistent BIM models of utility tunnels is proposed in this paper and it is applied to the modeling of the utility tunnel under Pearl Bay Plaza. The classification problem in complicated point cloud situations is solved with the proposed method of hierarchical segmentation by extracting walls, floors, ceilings, and the pipe system from point cloud blocks. Wall line features are extracted using an improved mean-shift algorithm, and bottom elevations of walls are extracted using Gaussian clustering. The mean-shift algorithm is also used to refine the pipe medial axis point set obtained by the medial axis extraction algorithm based on local symmetry, and the RANSAC method is used to extract segments of the pipe medial axes. Finally, a topologically consistent 3D model of the utility tunnel is reconstructed with the proposed topology reconstruction method by searching neighbor information of the wall and pipeline centerlines and establishing collinear, perpendicular, and intersecting scenarios. The effective application of this method verifies its reliability and provides a dependable solution for scan-to-BIM work in similar structures.

However, in this study, only the BIM modeling of underground utility tunnels is studied, and the modeling of aboveground buildings is not considered. Additionally, in-depth modeling of pipes, such as valves and flanges, was not performed in this study.

In our future work, pipeline components such as valves and flanges, will be identified from point clouds or pictures based on deep learning algorithms. Regarding the combination of modeling ground and underground building structures, we will use UAV images to reconstruct the ground buildings. Then, the ground and underground models will be incorporated into an integrated BIM model by aligning the model coordinates.

## Figures and Tables

**Figure 1 sensors-23-06503-f001:**
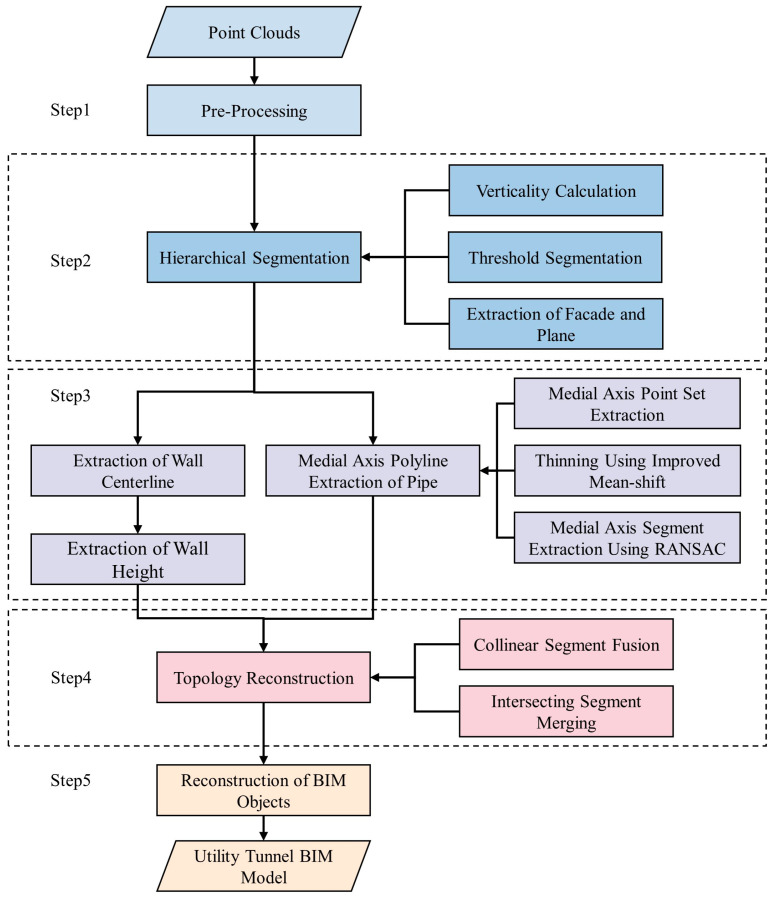
Workflow for generating topologically consistent BIM models of utility tunnels.

**Figure 2 sensors-23-06503-f002:**
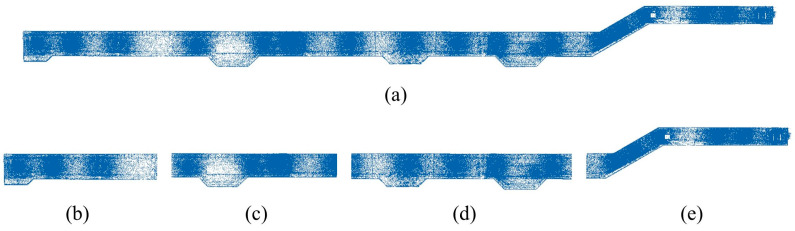
Illustration of the point cloud blocks. (**a**) Top view of the raw point cloud data P; (**b**–**e**) Top view of the four blocks partitioned from cloud P.

**Figure 3 sensors-23-06503-f003:**
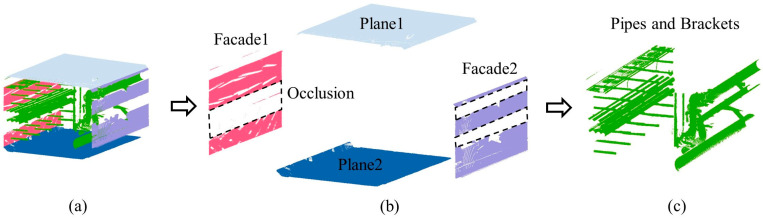
Illustration of the hierarchical segmentation method. (**a**) Different structures in the point cloud block are colored. (**b**) The clouds of the structures are extracted hierarchically. (**c**) The remaining pipes and brackets after extracting exterior structures.

**Figure 4 sensors-23-06503-f004:**
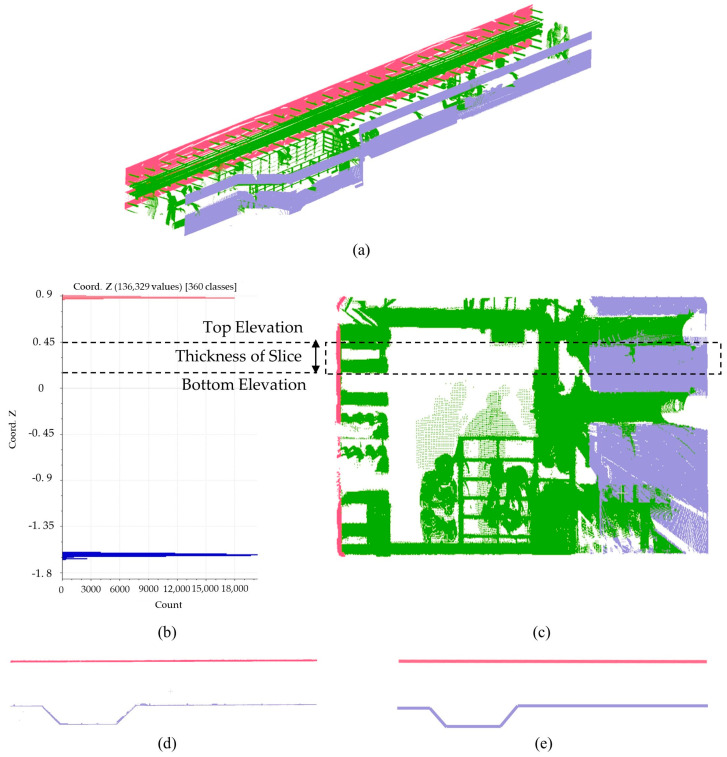
Illustration of obtaining slices from point cloud blocks and extracting wall lines. (**a**) A block after hierarchical segmentation; (**b**) Histogram of point cloud elevation and elevation range of the slice; (**c**) The specific position of the slice on the left view; (**d**) Wall clouds extracted from the slice; (**e**) Wall lines extracted from wall clouds.

**Figure 5 sensors-23-06503-f005:**
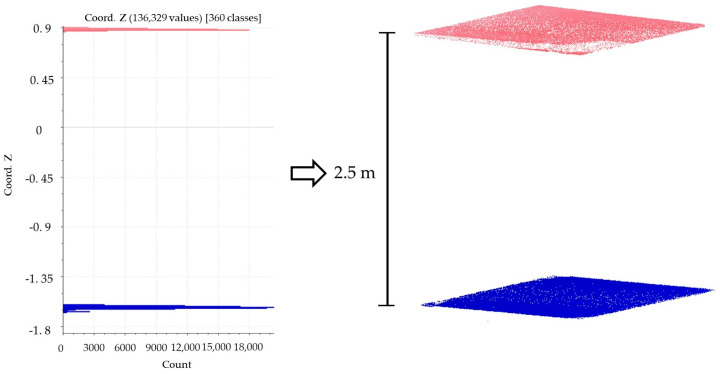
Extraction of the facade height.

**Figure 6 sensors-23-06503-f006:**
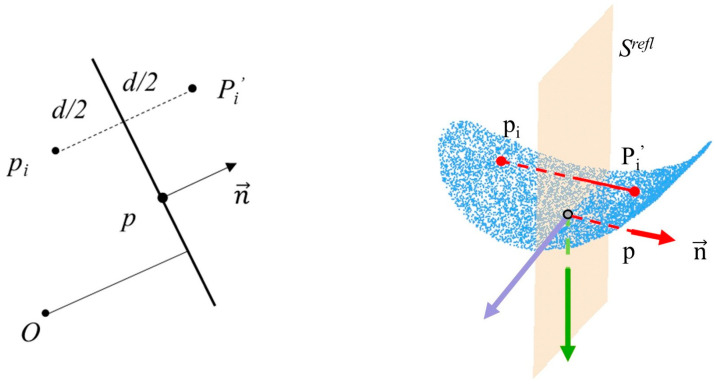
Mirror reflection symmetry analysis.

**Figure 7 sensors-23-06503-f007:**
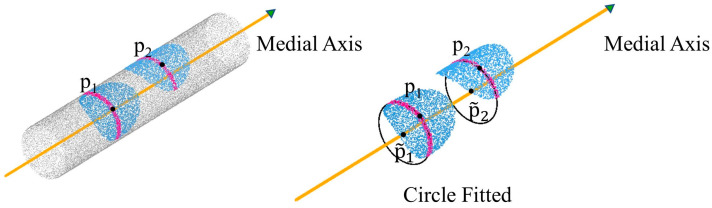
Using mirror reflection symmetry and RANSAC circle fitting to obtain the axis point set of the pipeline.

**Figure 8 sensors-23-06503-f008:**
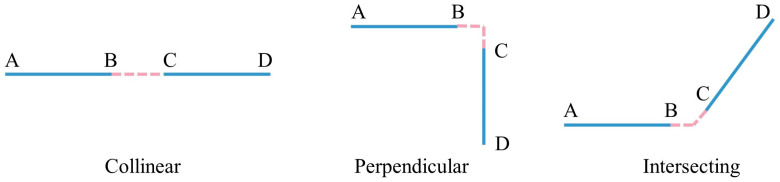
Three spatial relations involving line segments and their neighboring segments in Sets L and $L_{P}$.

**Figure 9 sensors-23-06503-f009:**
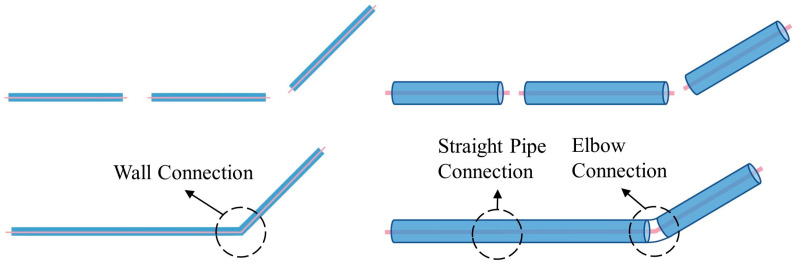
Topology reconstruction of the wall line and media axis of the pipes.

**Figure 10 sensors-23-06503-f010:**
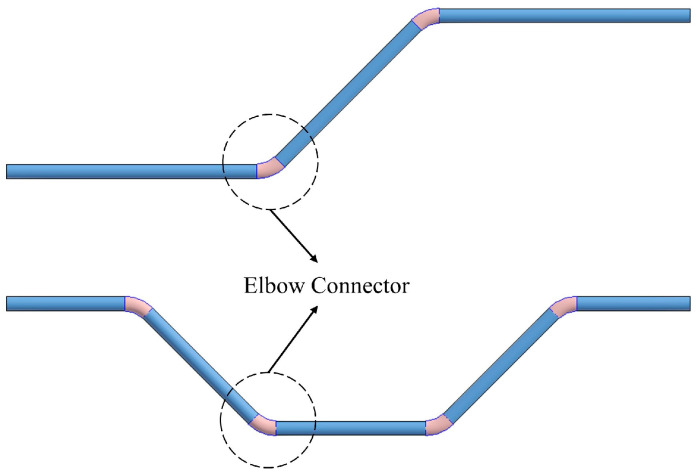
Pipe connectors for elbows.

**Figure 11 sensors-23-06503-f011:**
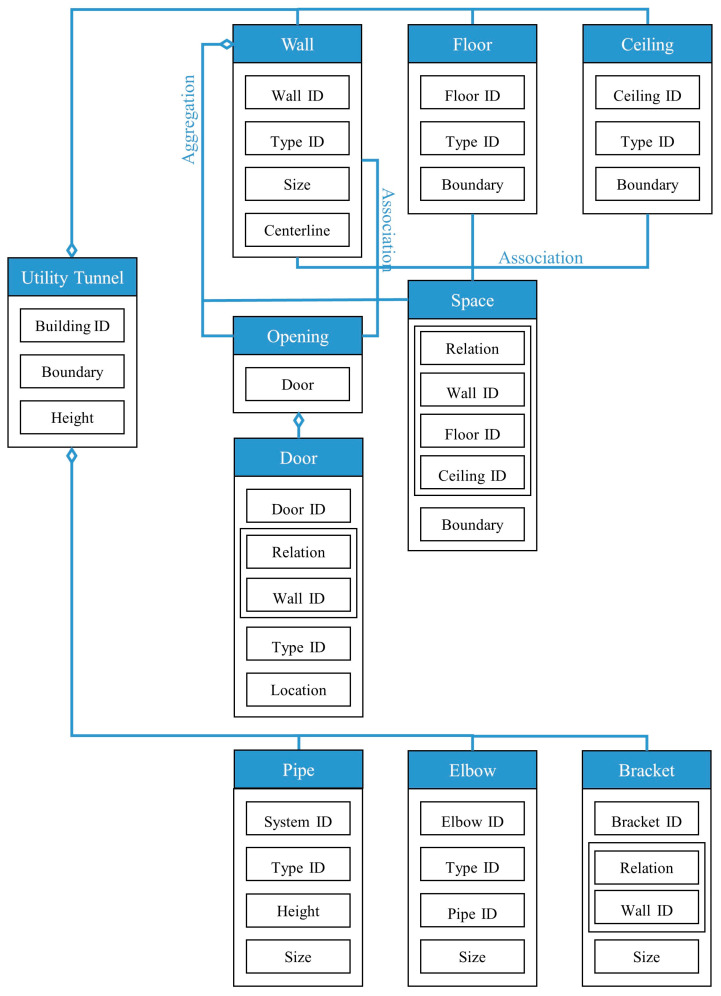
Representation and relationships of the parameters for the structures in the utility tunnel.

**Figure 12 sensors-23-06503-f012:**
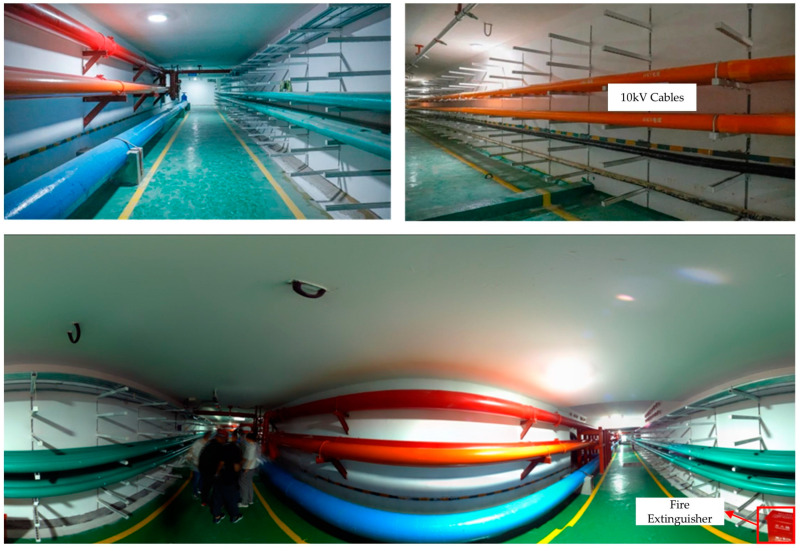
Pipeline distribution of the utility tunnel interior.

**Figure 13 sensors-23-06503-f013:**
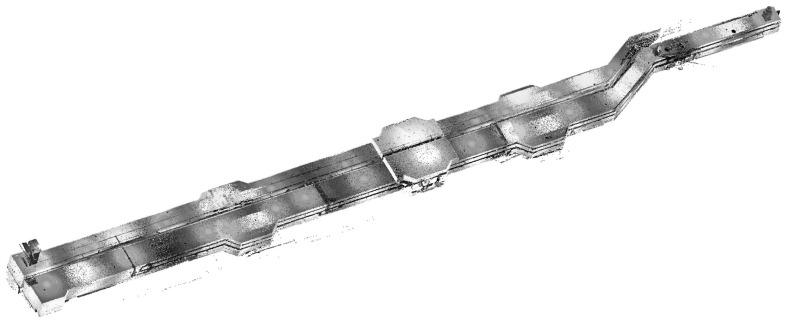
Measured point cloud of the utility tunnel under Pearl Bay Plaza.

**Figure 14 sensors-23-06503-f014:**
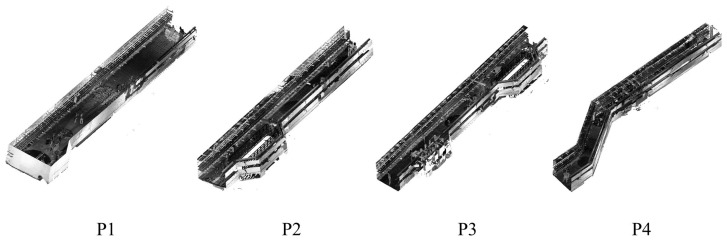
Measured point cloud of the utility tunnel under Pearl Bay Plaza.

**Figure 15 sensors-23-06503-f015:**
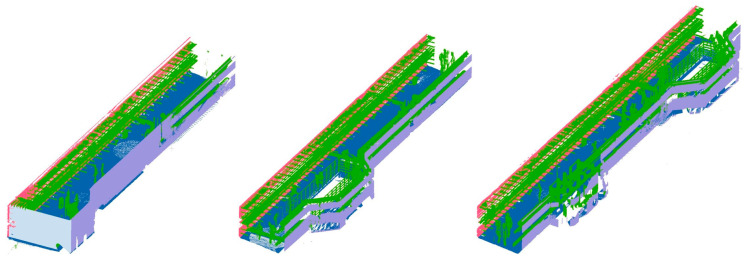
Hierarchical segmentation results of the point cloud block.

**Figure 16 sensors-23-06503-f016:**
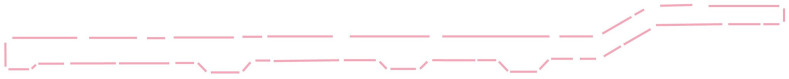
Extraction result of the wall line features.

**Figure 17 sensors-23-06503-f017:**
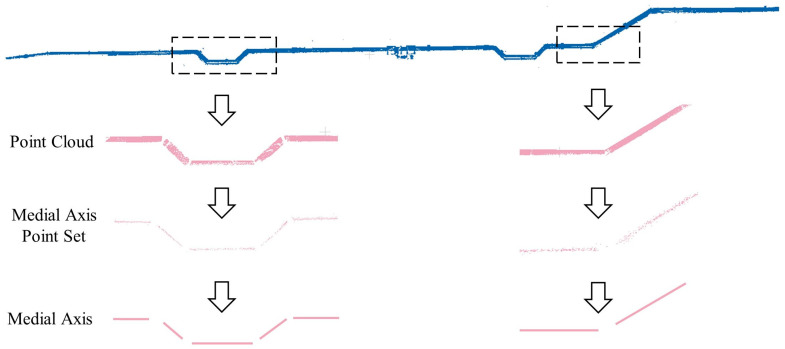
Extraction result of the medial axis segments of the pipe.

**Figure 18 sensors-23-06503-f018:**
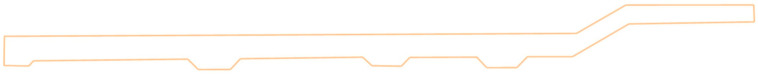
Exterior wall line features of the utility tunnel.

**Figure 19 sensors-23-06503-f019:**
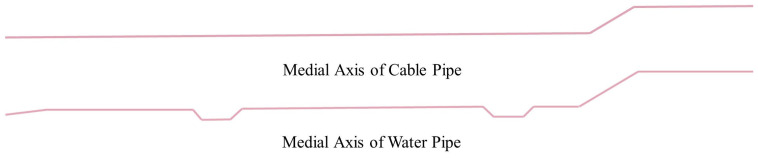
Complete medial axis of the pipes.

**Figure 20 sensors-23-06503-f020:**
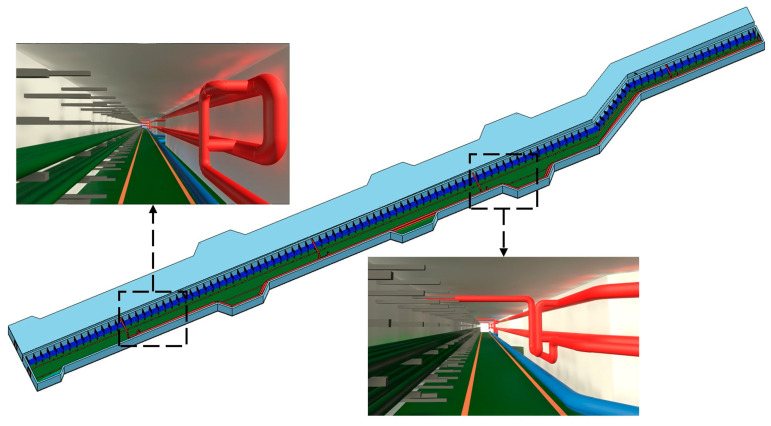
BIM model of the utility tunnel.

**Table 1 sensors-23-06503-t001:** Parametric structure examples.

Type	Sample	Parameters
Wall	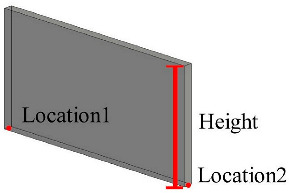	Wall ID: 1 Type ID: 332188 Size: Height 2500 mm Centerline: Location1, Location2
Door	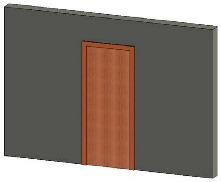	Door ID: 1 Relation: Embedding Wall ID: 1 Type ID: 332373
Floor	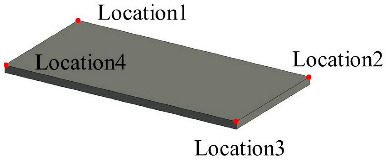	Floor ID: 1 Type ID: 332868 Boundary: Location1, Location2, Location3, Location4
Ceiling	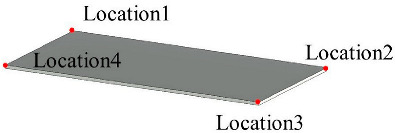	Floor ID: 1 Type ID: 332868 Boundary: Location1, Location2, Location3, Location4
Pipe	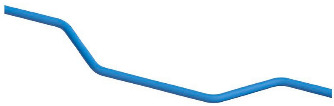	System ID: 743222 Type ID: 142438 Height: 240 mm Size: Diameter 250 mm
Bracket	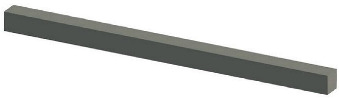	Bracket ID: 1 Relation: Fixing Wall ID: 1 Size: 900 mm × 50 mm × 50 mm

**Table 2 sensors-23-06503-t002:** Parameters of segmentations.

	Parameters	Segmentation Size			Points
ID		X/m	Y/m	Z/m	
P1		24.54	4.20	2.51	40,914,011
P2		24.68	4.98	2.53	53,182,745
P3		28.51	4.96	2.48	61,508,120
P4		30.38	6.96	2.46	72,103,920
Total		/	/	/	227,708,796

**Table 3 sensors-23-06503-t003:** Parameters of the BIM wall objects.

Wall Type	Count	Height/m	Thickness/ mm	Total Lenth/ m
Conventional-200 mm	24	2.5	200	223

**Table 4 sensors-23-06503-t004:** Examples and relative parameters of pipes.

Pipe System	Sample	Diameter	Lenth	Height	Count
Water	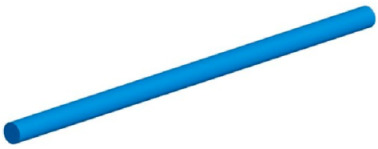	250 mm	112.7 m	240 mm	1
Fire	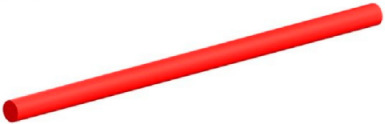	Main Pipe: 200 mm Branch Pipe: 150 mm	103.8 m 98.7 m 98.7 m	1100 mm 1125 mm 1875 mm	3
Cable	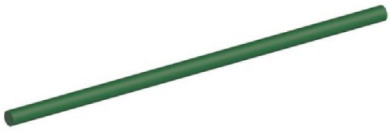	100 mm	110.8 m	850 mm 1150 mm	8

## Data Availability

Not applicable.

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
