# Peer review of "Generating Topologically Consistent BIM Models of Utility Tunnels from Point Clouds"

_sensors, 2023, doi:10.3390/s23146503_

Round 1

Reviewer 1 Report

The research presented in this paper provides a valuable solution for improving the accuracy of 3D models of utility tunnels by addressing the limitations of laser scanning technology. The proposed methods demonstrate efficiency and offer potential benefits for the planning, design, and maintenance of utility tunnels in municipal construction projects.

The abstract effectively summarizes the key points of the research.

Regarding Fig 2, it is crucial to improve its clarity and comprehensibility. The authors should consider revising the figure to enhance its visual representation and make it easier for readers to understand the content.

In relation to Fig 3, since its topic has been extensively examined in the literature, its inclusion may not be necessary. It is recommended to remove Fig 3 to streamline the presentation and avoid redundancy.

Overall, this research serves as a solid foundation for further advancements in the modeling of complex underground structures. It also highlights the potential for integrating underground and aboveground models, ultimately enhancing the overall representation and accuracy of building information models.

With the suggested revisions and improvements, the paper will provide a more refined and valuable contribution to the field.

Author Response

Dear Reviewer:

Thanks for your comments on our paper. We have carefully studied your comments and suggestions and made a correction that we hope will be approved. The revised portions are marked in red on the revised paper this time. Appended to this letter is our point-by-point response to your comments. Please see the attachment.

We appreciate your warm work earnestly and hope the corrections and responses will be approved. Again, thank you very much for your comments and suggestions.

Sincerely,

All authors

Reviewer 2 Report

Introduction The introduction is good, with some minor grammatical issues. It introduces the paper well and contextualizes the work. The inclusion of "the great urban disease" as a talking point in both the abstract and the introduction is questionable, though; I do not believe it adds to the paper and rather, is quite ambiguous and somewhat ominous in meaning. Consider omitting it.   Related Works
The related works section is well explained and covers tangentially related research while explaining where said research falls short the context of 3D mapping. Consider looking into the area of "search and rescue" for more on related works in this area.   Proposed Methods This section is well organized, and the sections allow for a clear breakdown in each step of the mapping method. The hierarchical segmentation approach, which allows various parts of the tunnels to be distinguished from one another is very novel and quite interesting. I think figure 5 should be clarified somewhat; it is not clear what the figure is illustrating at first glance.   Case Study This section explains a very good case study, in which the authors outline each step of applying their methods to a real world problem at Pearl Bay Plaza; minor fixes to the alignment of the figures should be performed, but the content is very good and concisely explained. Sections 4.6 and 4.7 have the same title. Renaming one or both is strongly advised for clarity and distinction.   Discussion Clear and concise, outlines the necessary takeaways from the case study.   Conclusion This section has a few minor grammatical errors; should be proofread more. Consider adding another paragraph to the end further detailing how you plan to tackle the issues outlined in the shortcomings of the study in this section.

The paper reads well and makes sense. There are small grammatical issues in various places. A moderate proofread is suggested in order for the English to read in a grammatically correct, coherent, tense-consistent fashion. Keep in mind that it is grammatically incorrect to switch from definite and indefinite articles within the same sentence, paragraph, etc. Also, keep an eye on consistency! Be consistent with words such as "façade", as you neglect to include the "ç" in some iterations of it. Small instances like this need to be fixed.

Author Response

(The authors gave the same response as above.)
